# Mechanical Properties of Human Concentrated Growth Factor (CGF) Membrane and the CGF Graft with Bone Morphogenetic Protein-2 (BMP-2) onto Periosteum of the Skull of Nude Mice

**DOI:** 10.3390/ijms222111331

**Published:** 2021-10-20

**Authors:** Md. Arafat Kabir, Akihiro Hirakawa, Bowen Zhu, Kenji Yokozeki, Mamata Shakya, Bingzhen Huang, Toshiyuki Akazawa, Masahiro Todoh, Masaru Murata

**Affiliations:** 1Division of Oral Regenerative Medicine, School of Dentistry, Health Sciences University of Hokkaido, Kanazawa 061-0293, Japan; kabir@hoku-iryo-u.ac.jp (M.A.K.); zhubear@hoku-iryo-u.ac.jp (B.Z.); yokozeki@hoku-iryo-u.ac.jp (K.Y.); shakyammta@gmail.com (M.S.); 2Biomechanical Design Faculty of Engineering, Hokkaido University, Kita 13, Nishi 8, Kita-ku, Sapporo 060-0819, Japan; akihiro1251@eis.hokudai.ac.jp (A.H.); todoh@eng.hokudai.ac.jp (M.T.); 3Corefront Corporation, 2-11 Ichitanihonmura-cho, Shinjuku-ku, Tokyo 162-0845, Japan; b.huang@corefront.com; 4Industrial Technology and Environment Research Development, Hokkaido Research Organization, Kita 19-jo Nishi 11-chome, Kita-ku, Sapporo 060-0819, Japan; akazawa-toshiyuki@hro.or.jp

**Keywords:** mechanical property, concentrated growth factor (CGF), BMP-2, bone induction, human

## Abstract

Concentrated growth factor (CGF) is 100% blood-derived, cross-linked fibrin glue with platelets and growth factors. Human CGF clot is transformed into membrane by a compression device, which has been widely used clinically. However, the mechanical properties of the CGF membranes have not been well characterized. The aims of this study were to measure the tensile strength of human CGF membrane and observe its behavior as a scaffold of BMP-2 in ectopic site over the skull. The tensile test of the full length was performed at the speed of 2mm/min. The CGF membrane (5 × 5 × 2 mm^3^) or the CGF/BMP-2 (1.0 μg) membrane was grafted onto the skull periosteum of nude mice (5-week-old, male), and harvested at 14 days after the graft. The appearance and size of the CGF membranes were almost same for 7 days by soaking at 4 °C in saline. The average values of the tensile strength at 0 day and 7 days were 0.24 MPa and 0.26 MPa, respectively. No significant differences of both the tensile strength and the elastic modulus were found among 0, 1, 3, and 7 days. Supra-periosteal bone induction was found at 14 days in the CGF/BMP-2, while the CGF alone did not induce bone. These results demonstrated that human CGF membrane could become a short-term, sticky fibrin scaffold for BMP-2, and might be preserved as auto-membranes for wound protection after the surgery.

## 1. Introduction

Regenerative medicine is based on applied biomaterials science. The normal wound healing process starts with blood coagulation consisting of the platelet aggregation and the formation of fibrin nets [1]. In 21st century, new platelet concentrates, so called platelet-rich fibrin (PRF) and concentrated growth factor (CGF), have been developed, and widely applied in clinical fields. Blood-derived materials from a patient can be used safety and immediately as autologous graft materials including growth factors.

Autologous platelet concentrates are generally divided into the following classifications: platelet-rich plasma (PRP), PRF, and CGF. PRP in the first generation is a liquid material including concentrated platelets in a small volume of plasma. As a disadvantage of PRP, a bovine blood-derived activator is needed for coagulation [2,3]. In 2006, PRF was developed in France by solely centrifugation without the addition of coagulation factors [4]. PRF has a solid fibrin matrix that contains a higher concentration of platelets, leucocytes, and growth factors compared to PRP [5]. Unlike PRF using a constant centrifugation speed, a machine for CGF switches the centrifugation speed automatically. CGF was developed by Sacco and published in 2010 [6]. CGF technology has interesting characteristics: i.e., the easy and speedy one-step preparation within 13 min, that switches the centrifugation speed to produce denser, richer fibrin matrix with growth factors [6]. CGF contained several growth factors including bone morphogenetic protein-2 (BMP-2) [7,8]. In our previous study, human CGF membrane combined with recombinant human BMP-2 (CGF/BMP-2) succeed bone induction at 14 days after the graft, though CGF alone did not induce bone in subcutaneous tissues [9]. Although commercially available collagen membranes are derived from pigs or cows, and dry materials without elasticity, the fresh CGF glue is 100% autologous and has wettability and elasticity. Until now, mechanical properties of human CGF have been not known well [10]. 

The ectopic studies of BMP-2 have been researched in rat subcutaneous tissues or rabbit muscles for more than 30 years since 1988 [11]. As a new graft site for BMP-2 bioassay, we have focused on a unique subcutaneous space between the periosteum and the galea of head, and bone induction was already confirmed by hydroxyapatite (HAp)/BMP-2 [12] or hydrogel/BMP-2 [13]. The supra-periosteal bone never united with skull [12,13], therefore, newly induced bone over original bone will become bone reservoir for transplantation. 

The aim of this study was to measure the tensile strength of human CGF membrane and to observe the behavior of human CGF membrane as a sticky scaffold of BMP-2 onto the periosteum under head skin of nude mice.

## 2. Results

### 2.1. Histological and SEM Findings of Concentrated Growth Factor (CGF) Glue

The CGF glue showed mainly yellow, accompanied with a small red part (Figure 1a). Hematoxylin and eosin (HE)-stained whole view consisted of three parts (mainly week pinkish, thin layer mixed with violet and red, red clot) as shown in Figure 1b. Scanning electron microscopy (SEM) of the yellow glue part in Figure 1a showed multiple oblong platelets on the fibrin strand (Figure 1c). 

### 2.2. Initial Length, Width, and Thickness of CGF Membranes

Sequential length, width, and thickness of the test membranes were summarized in Table 1. The appearance and size of the samples were almost same for seven days by soaking at 4 °C in saline. 

### 2.3. Tensile Strength and Elastic Moduli

The tensile strength (obtained from the maximum stress) and elastic modulus (obtained from the linear region of each stress-strain curve) are shown in Figure 2a,b, respectively. The tensile strength was in the range of 0.10 to 0.44 MPa. The elasticity in the range of strain from 20% to 30% was in the range of 0.10 to 0.26 MPa. The elasticity at 0 day to the tensile strength was in the range of 0.10 to 0.35 MPa. No significant differences of both the tensile strength and the elastic modulus were found among the different preservation period. 

### 2.4. Concentrated Growth Factor with Recombinant Human Bone Morphogenetic Protein-2 (CGF/BMP-2) Membrane Onto Periosteum of Skull of Head Skins

Supra-periosteal bone induction occurred at 14 days. The newly induced bone mass was over the skull and the periosteum existed between new bone and the skull (Figure 3a). Undifferentiated mesenchymal cells proliferated inside C-shaped induced bone (Figure 3b), revealing woven bone structure, not lamellar (Figure 3c). Monocytes and lymphocytes appeared slightly in the surrounding fibrous tissues of induced bone, but acute inflammation was never observed at 14 days. Ectopic bone was divided from the skull by fibrous connective tissues including periosteum. Weekly hematoxylin-stained matrix and eosin-stained matrix were observed continuously in induced bone (Figure 3d). Hypertrophic chondrocyte-like cells existed in the hematoxylin-stained matrix (Figure 3e). The CGF membrane was totally replaced by bone and mesenchymal tissues at 14 days. Hypertrophic chondrocyte-like cells existed in the hematoxylin-stained matrix (Figure 3e). The CGF membrane was totally replaced by bone and mesenchymal tissues at 14 days.

In the other specimen block as shown in Figure 4, circular frame bone was induced and almost all CGF disappeared (Figure 4a). A young osteocyte and cuboidal osteoblasts were found nearby strongly eosin-stained acellular CGF bundle. Induced bone matrix was stained weekly by eosin less than the residue of CGF bundle (Figure 4b). The bone structure didn’t show lamellar. A part of induced bone showed cartilage-like appearance (Figure 4c). Immature bone with cuboidal osteoblast lining and capillaries (Figure 4d).

### 2.5. Concentrated Growth Factor (CGF) Membrane Alone onto Periosteum of Skull of Head Skins

The CGF alone did not induce bone. Undifferentiated mesenchymal cells proliferated, and fibroblast differentiation were observed between thick CGF bundles at 14 days. The bundle body appeared strongly eosin-stained acellular matrix. Monocytes and lymphocytes appeared slightly between the residues of CGF. The membrane surface represented irregular without the encapsulation of fibrous tissues, and several giant cells were observed at 14 days (Figure 5). 

## 3. Discussion

Mechanical properties of the full length of human CGF membranes were assessed at 0, 1, 3, 7 days in this study. The tensile strength of the full length was 0.24 MPa, 0.27 MPa at 0 day, 7 days, respectively. The strength and elastic modulus did not decrease significantly over time, indicating that the mechanical properties could be preserved in 0.9% NaCl at 4 °C for 7 days. The previous paper reported the tensile strength of fresh CGF (length:10 mm) was 0.17 MPa at a stretching speed of 1 mm/min [10]. Our test method utilized the full length (average: 27.4 mm) and set at a stretching speed of 2 mm/min. Though the test methods were different, the tensile strength of the full length (0.24 MPa) was similar to that of 10 mm-membrane (0.17 MPa). Especially after bone augmentation in implant surgery, wound dehiscence accidentally occurs a few days later. If patient-owned CGF membranes were stocked in 0.9% NaCl at 4℃ after the surgery, doctors could cover the exposed tissues by the sticky membrane with original mechanical property. We believe, therefore, CGF might be applied for not only the cure surgery, but also the post-operative care. Though the tensile test was important in evaluating the mechanical properties of CGF membranes, there was difficulties related with the chucking method for the test pieces, and some data could not be obtained for broken pieces. The present extra-skeletal study showed a unique bone induction by human CGF/BMP-2. Regarding the choice of animal, nude mouse is ideal for assessing human-derived materials, as the immunocompromised animals do not reject human graft materials. In our study, nude mice received human biological materials, and monocytes and lymphocytes appeared just a little in both CGF/BMP-2 and CGF alone. We therefore believe the findings mean the rejection and the strong inflammation did not occur in the body of nude mouse.

BMP-2 requires biomaterials, which function as sustained-release carriers at the graft site. The CGF/BMP-2 induced bone at 14 days subcutaneously over the skull. The induced bone showed egg-shell-like shape, and the fresh CGF membrane was gradually absorbed, and replaced by bone and fibrous connective tissues until 14 days. The extra-skeletal head skin tissue was selected for bioassay in this study, because mice did not have volume of muscles that receiving human CGF membrane (5 × 5 × 2 mm^3^). From a biological point of views, the circumstance of muscle pouch has a better performance in bone induction than that of skin tissue due to the number of capillaries. It is well known that BMP-2 targets undifferentiated mesenchymal cells nearby capillaries. On the other hand, CGF alone failed to bone induction as previously reported [9]. Our results indicated that the growth factors in CGF alone could stimulate the proliferation of mesenchymal cells, but they do not have the ability to differentiate undifferentiated mesenchymal cells into osteogenic cells. The histological findings in CGF alone were consistent with the biochemical report that the concentration of BMP-2 in CGF was almost undetectable by ELISA [14]. Recently, synergistic activity of TGF-β1/BMP-2 or VEGF/TGF-β1/BMP-2 was reported for osteoblast differentiation [15,16]. The in vitro reports may support the synergistic effects of BMP-2 and CGF-released growth factors for bone induction in this bioassay. 

Collagen-based materials are extremely important for regenerative medicine, because collagen is a main organic component in human body. Nowadays, atelocollagen materials are widely applied for surgery and in the post-operative care processes [17,18,19]. However, we cannot produce a patient-derived collagen immediately in clinics, and commercially available collagenous materials do not include any growth factors. Moreover, Muslims reject pig-derived collagen materials. In contrast, CGF is 100% autologous blood products, composed of cross-linked fibrin strands and various growth factors (PDGF, VEGF, IGFs and TGF-β1) [20], and can be prepared immediately in clinics by ourselves for surgery. In addition, it is possible to fabricate various shapes by doctor’s hand, and CGF is free of the risk of cross-contamination due to 100% patient-own blood. As for bio-absorption, the cross-linked fibrin strands are digested by plasmin [9,21]. Collagen is resistant to the proteolysis for a triple helix structure, although single-stranded regions are cleaved by matrix metalloproteinases (MMPs) [22].

Collagen/BMP-2 composite has been commercially available for clinical use in USA and EU [23], but not in Japan. In the near future, the combined use of wet CGF and dry collagen will be applied clinically as a novel delivery scaffold for BMP-2.

## 4. Materials and Methods

### 4.1. Preparation of Concentrated Growth Factor (CGF) Glue

Blood samples were collected from four non-smoking, healthy, male volunteers with ages ranging from 25 to 35 years. The donors had no hindrance in daily life, nor did they have any systemic diseases. Sixty-four milliliters of peripheral venous blood were collected by using 18G needle in eight sterilized vacuum glass tubes without anticoagulant additives (Figure 6a). The tubes were immediately centrifuged in a special centrifuge device (Medifuge^Ⓡ^, Silfradent Srl, Forli, Italy) with the following automatic programs: 30 s acceleration, 2 min at 2700 rpm, 4 min at 2400 rpm, 4 min at 2700 rpm, 3 min at 3000 rpm, and 36 s deceleration and stop (Figure 6b). After the systematized centrifugation, the upper layer (platelet-poor plasma, PPP) was discarded (Figure 6c), and the middle layer (yellow glue with buffy coat) was collected with sterile tweezers as CGF glue (Figure 6d) for histological and scanning electron microscopy (SEM) observations. 

#### 4.1.1. Histological Observation of Fresh CGF Glue

Fresh CGF glue was fixed in a 10% neutral phosphate-buffered formalin solution for 24 h, dehydrated with ascending grades of ethanol (50–100%), and processed for paraffin embedding. Later, histological sections (thickness: 5 µm) were prepared and stained with hematoxylin and eosin (HE).

#### 4.1.2. Scanning Electron Microscopy (SEM) Observation of Fresh CGF Glue

SEM (JSM-6610LA^Ⓡ^, Jeol, Tokyo, Japan) was used to investigate the microstructures of the CGF glue. The samples were fixed with 2% neutralized glutaraldehyde for 1 h, and dehydrated serially in 30%, 50%, 70%, 90%, and 100% ethanol solutions. The SEM procedures were completed by critical drying point of the material. Finally, the samples were observed, and the representative images were captured by SEM with an accelerating voltage of 15 kV. 

### 4.2. Mechanical test of Concentrated Growth Factor (CGF) Membrane

#### 4.2.1. Preparations of Concentrated Growth Factor (CGF) Membrane 

The CGF glues were collected from the same 4 volunteers as shown in Figure 6 and pressed by a stainless-steel compression device (Sticky Bone TM, Seoul, Jeonju, Korea) to prepare the CGF membranes (Figure 7a–c). For ectopic bioassay, the pressed CGF was transformed into a graft membrane (5 × 5 × 2 mm^3^).

#### 4.2.2. Mechanical Properties of Concentrated Growth Factor (CGF) Membrane

Eight CGF membranes were obtained from a volunteer and were divided each two into 4 groups (0 days, 1 day, 3 days, and 7 days after the preparation of CGF glues) for the tensile tests. Each group includes 8 test membranes. Tensile tests were performed using a universal testing machine (Model3365, Instron Corp., Norwood, MA, USA), equipped with a 50N uniaxial load cell. Both ends of CGF membranes were fixed to the heads of the universal testing machine binding between the metal jig and the acrylic plate (Figure 7d) and captured images by a digital camera from the frontal side (Figure 7d) and the lateral side (Figure 7e) at the non-strained state for calculating the cross-sectional area. The strain of each CGF membrane was estimated from the distance of both heads. All the tensile tests for CGF membranes were carried out at constant strain rate of 2.0 mm/min until breakage at room temperature. The elastic modulus and ultimate strength for each CGF membrane were then obtained. The elastic modulus was then calculated from the linear region (strain from 20% to 30%) of stress-strain curve of each CGF membrane. The tensile strength was determined as the maximal stress under tensile loading. All test membranes were preserved in saline at 4 °C.

### 4.3. Animal Experiments

#### 4.3.1. Preparation of Concentrated Growth Factor with Recombinant Human Bone Morphogenetic Protein 2 (CGF/BMP-2) Membranes

Forty microliters of BMP-2 solution (0.025 g/L; Astellas Pharma Inc., Tokyo, Japan) were added to the CGF membrane (5 × 5 × 2 mm^3^). The dose of BMP-2 was based on a dose-dependent study in subcutaneous tissues, and 1.0 μg of BMP-2 per carrier (5 × 5 × 5 mm^3^) was considered a relatively critical dose for bone induction in subcutaneous tissues [24]. The mixture was kept at 4 °C for 1 h until in vivo study. As a control, 40 µL of distilled water was added to the CGF membrane (5 × 5 × 2 mm^3^).

#### 4.3.2. Ectopic Bioassay (Supra-Periosteal Site)

Ten nude mice (5-week-old, male) were used for the assessment of in vivo ectopic bone formation. Ketamine hydrochloride (0.1 mg/kg body weight, Ketalar^®^ 50 mg/mL; Daiichi Sankyo Propharma Co., Ltd., Tokyo, Japan) and xylazine hydrochloride (0.01 mg/kg body weight, Skill pen^®^ 20mg/mL; Interbet Co., Ltd., Tokyo, Japan), diluted with 0.9% NaCl, were used as anesthetics for general anesthesia. Each mouse had one horizontal skin incision on head and the galea on the periosteum was elevated for the graft. The CGF/BMP-2 membrane or the CGF alone was grafted onto the periosteum (Figure 8), and the head skin was repositioned and sutured. Management of post-operative pain included subcutaneous administration of buprenorphine. The health and behavior of the mice was monitored twice a week in the first week and once a week thereafter. At 14 days after the graft, mice were euthanized with an overdose of anesthesia, and the specimen was harvested with skull (Figure 8).

#### 4.3.3. Tissue Preparation 

The specimens were fixed in a 10% neutral phosphate-buffered formalin solution for 3 days, demineralized in 10% formic acid for 10 days, and processed for paraffin embedding. Cross sections (thickness: 5 µm) were prepared for hematoxylin and eosin (HE) staining. 

### 4.4. Ethical Approval

The study design and the consent forms for the procedures performed with the study subjects were approved (Code no.150, issued on May 2018) by the ethical committee for human subjects at Health Sciences University of Hokkaido with the principles of the Declaration of Helsinki. Blood samples were taken from donor volunteers after obtaining verbal consent. All animal bioassays in this study were conducted in accordance with the Institutional Animal Care and Use Committee of Health Sciences University of Hokkaido.

## 5. Conclusions

The CGF/BMP-2 membrane induced bone with cartilage-like tissue at 14 days. The CGF membrane alone could not induce bone and almost absorbed under head skin at 14 days. The CGF membrane could support the capability of BMP-2 as the absorbable and sticky fibrin matrix. The appearance and size of the CGF membranes were almost same for 7 days by soaking at 4 °C in saline, and the tensile strength could be preserved for 7 days. These results demonstrated that human CGF membrane could act as a biological scaffold for the delivery of BMP-2 and be reserved for wound healing after the surgery. The CGF/BMP-2 might become bone reservoir on the periosteum of skull for bone autograft.

## Figures and Tables

**Figure 1 ijms-22-11331-f001:**
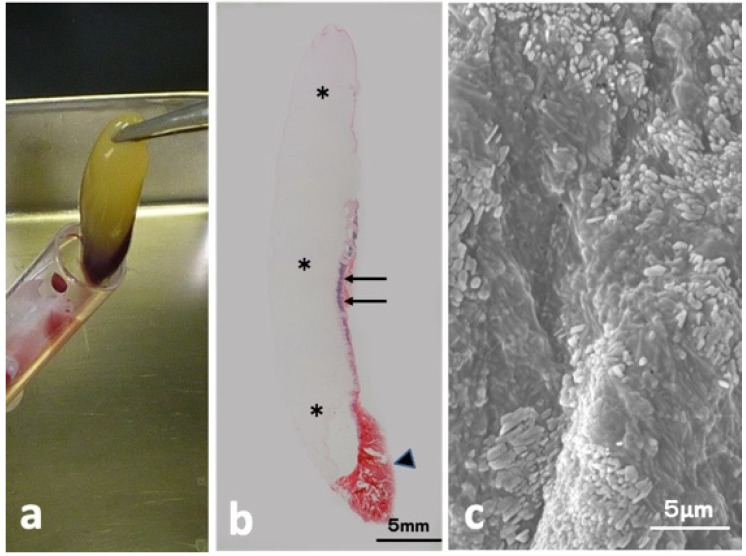
CGF glue. (**a**) Gross view of CGF, (**b**) HE-stained whole view showing week pinkish portion (*), violet (arrow), and red (arrowhead) (HE), (**c**) SEM showing oblong-shaped platelets on fibrin matrices.

**Figure 2 ijms-22-11331-f002:**
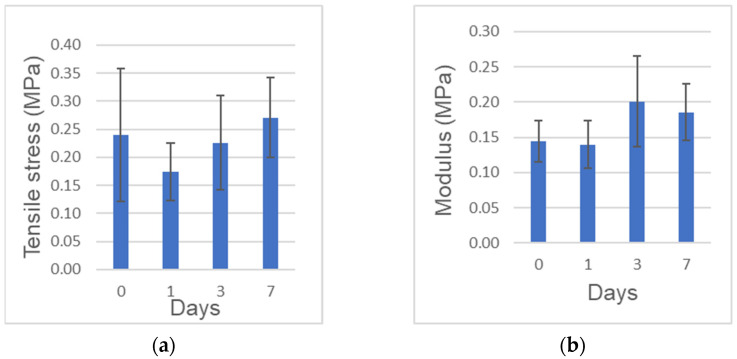
Mechanical tests. (**a**) Tensile strength. (**b**) Elastic modulus at 20–30%. *n* = 6–7.

**Figure 3 ijms-22-11331-f003:**
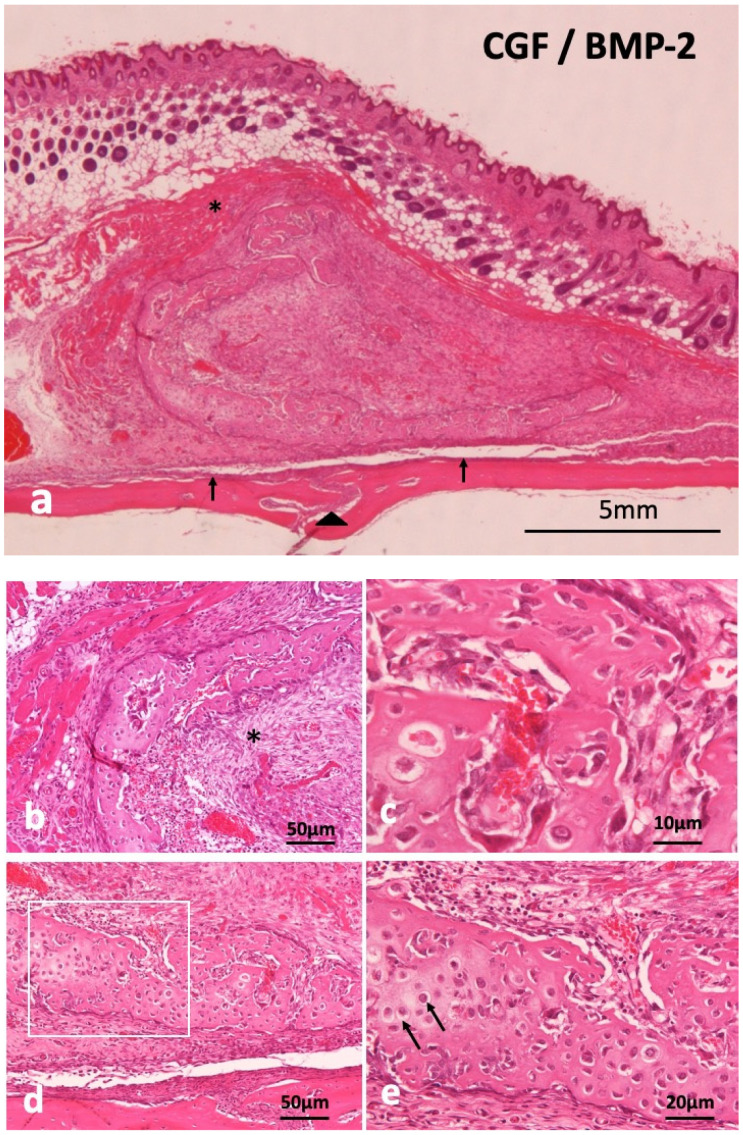
Histological images of CGF/BMP-2 membrane on periosteum of skull at 14 days. (**a**) Induced bone between muscle and periosteum, arrowhead indicating middle suture part of skull, (**b**) Trabecular bone and undifferentiated mesenchymal cells (*), (**c**) Woven bone revealing wavy surface structure and irregular arrangement of osteocytes, (**d**) Mix of weekly hematoxylin-stained matrix and eosin-stained matrix in induced bone, (**e**) Higher magnification of square in (**c**). Arrow indicating round shaped- hypertrophic chondrocyte-like cells (HE).

**Figure 4 ijms-22-11331-f004:**
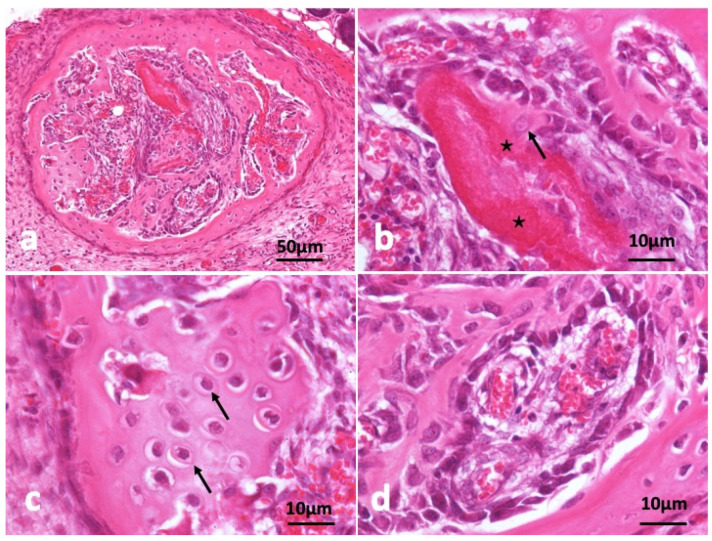
Histological images of CGF/BMP-2 membrane on periosteum of skull at 14 days. (**a**) CGF/BMP-2 induced bone like eggshell, (**b**) Osteoblasts near residue of strongly eosin-stained CGF (*) representing acellular, arrow indicating young osteocyte, (**c**) Arrow indicating osteocytes like hypertrophic chondrocytes, (**d**) Immature bone with cuboidal osteoblast lining near capillary (HE).

**Figure 5 ijms-22-11331-f005:**
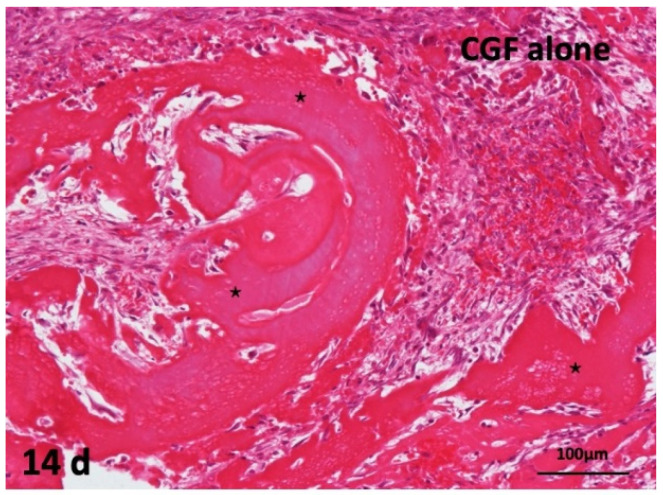
Histological images of CGF alone membrane on periosteum of skull. Acellular appearance inside CGF bundle (*) and cellular invasion into spaces between bundles (HE).

**Figure 6 ijms-22-11331-f006:**
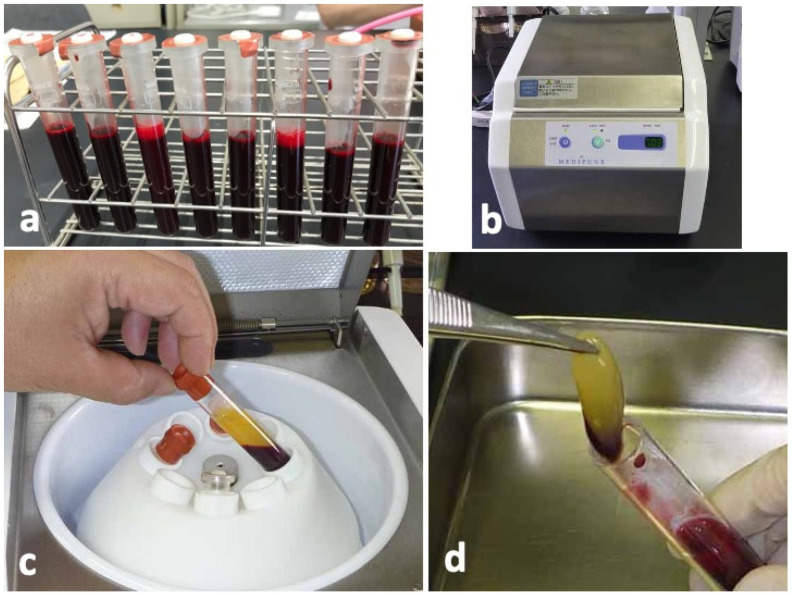
Preparation of concentrated growth factor (CGF) glue. (**a**) Venous blood in vacuum glass tubes, (**b**) Centrifuge machine (Medifuge^Ⓡ^, Silfradent Srl, Forli, Italy) used for preparation of CGF glue, (**c**) Blood fractions just after centrifugation, (**d**) Middle layer (CGF glue) showing yellow glue with a small amount of red layer (buffy coat).

**Figure 7 ijms-22-11331-f007:**
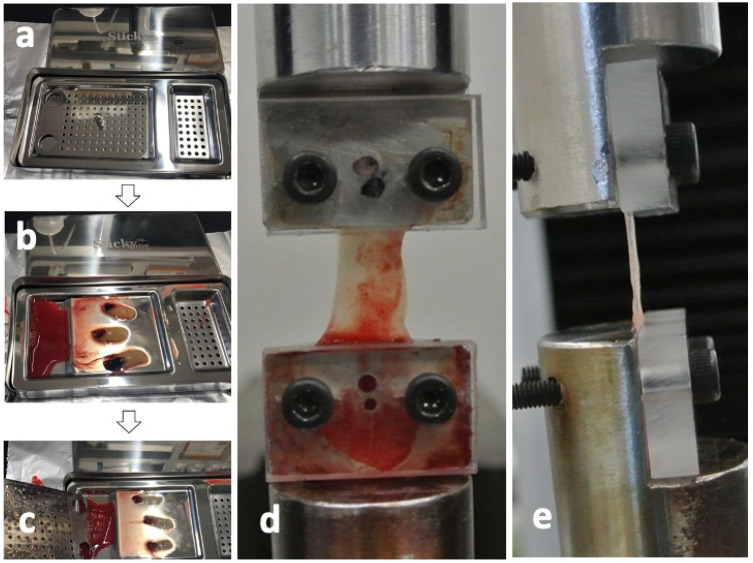
Test preparation of concentrated growth factor (CGF) membrane. (**a**) Stainless-steel compression device (Sticky Bone^TM^, Jeonju, Korea) to transform into CGF glue into membrane, (**b**) CGF glue on device before press, (**c**) CGF membrane after press, (**d**) frontal view after CGF set, (**e**) lateral view after CGF set.

**Figure 8 ijms-22-11331-f008:**
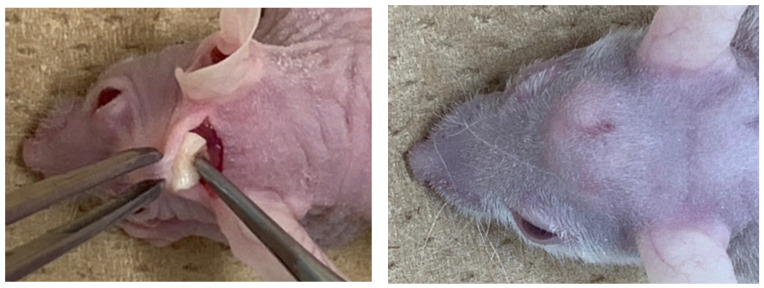
Gross views at CGF graft surgery (**left**) and dome-like uplift at 14 days (**right**).

**Table 1 ijms-22-11331-t001:** Size of CGF membrane at each day.

Day	Length (mm)	Width (mm)	Thickness (mm)
0	27.26 ± 1.38	7.27 ± 1.19	0.58 ± 0.10
1	28.92 ± 3.55	6.41 ± 0.74	0.58 ± 0.09
3	28.80 ± 1.59	7.13 ± 0.64	0.43 ± 0.07
7	26.02 ± 1.89	6.89 ± 0.80	0.45 ± 0.07

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
