# Peer review of "Mechanical Properties of Human Concentrated Growth Factor (CGF) Membrane and the CGF Graft with Bone Morphogenetic Protein-2 (BMP-2) onto Periosteum of the Skull of Nude Mice"

_ijms, 2021, doi:10.3390/ijms222111331_

Round 1

Reviewer 1 Report

This paper shows that human CGF membranes can serve as short-term adhesive fibrin scaffolds for BMP-2 and can be preserved for wound protection after surgery. It is of great clinical interest.
In section 2.2, the authors should add an explanation of how they preserved the CGF membranes for 7 days.

Author Response

  1. Next sentence was added in Abstract and Section 2.2.

Appearance and size of samples were almost same by soaking at 4℃ for 7 days in saline. 

  1. Next sentence was added in Conclusion

The appearance and size of the CGF membranes were almost same for 7 days by soaking at 4℃ in saline, and the tensile strength could be preservedfor 7 days.

Reviewer 2 Report

This manuscript presents very interesting data on applied biomaterials science that are clearly of broad interest and deserve publication. However, it needs some correction.

In results and discussion,

Please describe the difference in reactive inflammation of the graft site between CGF/BMP-2 and CGF alone.

Figure 1b: Please add a scale bar.

Figure 8: The photo is a little bit blurry, so please replace it with a clearer one.

Author Response

1-1. Next sentence was added in result of CGF/BMP-2.

Monocytes and lymphocytes appeared slightly in the surrounding fibrous tissues of induced bone, but acute inflammation was never observed at 14 days.

1-2. Next sentence was added in result of CGF alone.

Monocytes and lymphocytes appeared slightly between the residues of CGF.

1-3. Next sentence was added in Discussion.

Nude mice received human biological materials, and monocytes and lymphocytes appeared just a little in both CGF/BMP-2 and CGF alone. We believe, therefore, the findings mean the rejection and the strong inflammation didn’t occur in the body.

  1. Scale bar was added in Figure 1b.
  2. Clear photos were replaced in Figure 8.
